# Habits, Goals, and Behavioral Signs of Cognitive Perseveration on Wisconsin Card-Sorting Tasks

**DOI:** 10.3390/brainsci13060919

**Published:** 2023-06-06

**Authors:** Bruno Kopp, Bilal Al-Hafez, Alexander Steinke

**Affiliations:** Department of Neurology, Hannover Medical School, Carl-Neuberg-Straße 1, 30625 Hannover, Germany

**Keywords:** Wisconsin card sorting, cognitive flexibility, cognitive perseveration, error suppression, instrumental learning, goal-directed control, GIC

## Abstract

Wisconsin card-sorting tasks provide unique opportunities to study cognitive flexibility and its limitations, which express themselves behaviorally as perseverative errors (PE). PE refer to those behavioral errors on Wisconsin card-sorting tasks that are committed when cognitive rules are maintained even though recently received outcomes demand to switch to other rules (i.e., cognitive perseveration). We explored error-suppression effects (ESE) across three Wisconsin card-sorting studies. ESE refer to the phenomenon that PE are reduced on repetitive trials compared to non-repetitive trials. We replicated ESE in all three Wisconsin card-sorting studies. Study 1 revealed that non-associative accounts of ESE, in particular the idea that cognitive inhibition may account for them, are not tenable. Study 2 suggested that models of instrumental learning are among the most promising associative accounts of ESE. Instrumental learning comprises goal-directed control and the formation of corresponding associative memories over and above the formation of habitual memories according to dual-process models of instrumental learning. Study 3 showed that cognitive, rather than motor, representations of responses should be conceptualized as elements entering goal-directed instrumental memories. Collectively, the results imply that ESE on Wisconsin card-sorting tasks are not only a highly replicable phenomenon, but they also indicate that ESE provide an opportunity to study cognitive mechanisms of goal-directed instrumental control. Based on the reported data, we present a novel theory of cognitive perseveration (i.e., the ‘*g*oal-directed *i*nstrumental *c*ontrol’ GIC model), which is outlined in the Concluding Discussion.

## 1. Introduction

The Wisconsin card-sorting task [1,2] provides one of the most popular neuropsychological assessment techniques for executive functioning [3,4,5]. The Wisconsin card-sorting task examines individual abilities to form abstract concepts, and to maintain or shift mental sets in response to verifying or falsifying feedback [6,7,8], as detailed in Figure 1.

There is no explicit mentioning of the to-be prioritized sorting rule on Wisconsin card-sorting tasks. Rather than that, examinees need to infer the currently prioritized rule from verifying or falsifying feedback, which the examiner provides on a trial-by-trial basis. Verifying feedback is often expressed by the word ‘CORRECT’, which signifies that the correct rule has been applied on the current trial. In the present study, verifying feedback was expressed by the word ‘REPEAT’, which signified that the currently applied rule could be maintained on the upcoming trial [11]. Falsifying feedback is often expressed by the word ‘INCORRECT’, which signifies that the incorrect rule has been applied on the current trial. In the present study, falsifying feedback was expressed by the word ‘SWITCH’, which signified that a rule switch was requested on the upcoming trial [11].

Performance on Wisconsin card-sorting tasks provides behavioral indicators of cognitive perseveration [12]. Figure 1A illustrates how cognitive perseveration may generate a specific type of behavioral errors on Wisconsin card-sorting tasks across two successive trials. In the depicted examples, SWITCH-feedback on trial *t* − 1 requested switching away on trial *t* from the COLOR-rule that had been executed on all exemplified *t* − 1 trials. Re-executions of the COLOR-rule on trial *t* represent perseverative errors (PE; potential PE are highlighted as red response buttons on exemplified trials *t*). PE are behavioral signs of cognitive perseveration since they indicate that the previously prioritized sorting rule was re-executed despite the fact that the occurrence of the SWITCH-feedback on trial *t* − 1 signaled the need to prioritize another sorting rule on trial *t* (i.e., either the SHAPE-rule or the NUMBER-rule on the exemplified trials).

Our group recently discovered error-suppression effects (ESE; see Figure 1 for illustration). As shown in Figure 1B,C, ESE refer to the phenomenon that conditional probabilities of PE on repetitive trials (repeated key cards incl. features and responses) are lower than conditional probabilities of PE on non-repetitive trials (altered key cards incl. features and responses). We refer to the former PE probabilities as repetitive PE and to the latter PE probabilities as non-repetitive PE. The term repetitive signifies the repetition of key cards/features and responses across two successive trials (see Figure 1A, right panel). Note that only repetitive PE could be conceived as behavioral signs of motor perseveration [12]. The term non-repetitive signifies altered key cards/features and responses across two successive trials (see Figure 1A, left panel). Note that non-repetitive PE cannot be conceived as behavioral signs of motor perseveration [12]. (Note that we change our terminology at this point. Key card (feature)/response alterations were actually referred to as ‘demanded response repetitions’ (meaning that preventing the occurrence of perseverative errors demanded response repetitions) in our previous ESE publications [9,10,13]. Key card (feature)/response repetitions were then referred to as ‘demanded response alterations’ (meaning that preventing the occurrence of perseverative errors demanded response alterations). These notational changes may look a bit confusing, but the present terminology is much more intelligible. For the sake of simplicity, utilizing the novel terminology is highly recommended in future studies.) ESE simply express the phenomenon that repetitive PE are reduced compared to non-repetitive PE. Our previous studies showed that ESE are a well-replicable phenomenon on Wisconsin card-sorting tasks [9,10,13].

The present series of three studies aims at investigating explanatory concepts for ESE. Study 1 examined contributions from conceivable non-associative and associative accounts of the origin of ESE. The result of Study 1 was that ESE depend on conjunctive associations between key-card features and responses. Study 2 examined contributions from classical and instrumental learning accounts of ESE. The result of Study 2 was that theories of instrumental learning provide the more efficient explanatory concept for ESE. More specifically, dual-process models envisage instrumental learning as the formation of associative memories at two separate levels [14,15,16]. The habitual level of instrumental learning presumes the formation of associations between stimuli and responses. The goal-directed level of instrumental learning and the formation of corresponding memories remains contingent upon action–outcome monitoring. The main conclusion from Study 2 was that ESE on Wisconsin card-sorting tasks result from shifts in the balance between habitual and goal-directed instrumental learning, with stronger goal-directed control on repetitive trials than on non-repetitive trials: repetitive PE are reduced compared to non-repetitive PE due to the preponderance of goal-directed control on repetitive trials. Study 3 finally showed that cognitive, rather than motor, representations of responses constitute elements of goal-directed instrumental learning on Wisconsin card-sorting tasks.

## 2. General Method

### 2.1. Participants

Participants were recruited via email and messaging platforms from the student populations of the Hannover Medical School and Leibniz University Hannover. They were enrolled in medicine, biochemistry, and electrical and mechanical engineering. No psychology students were recruited since these two universities do not offer a psychology major. None of the participants had ever attended neuropsychology lectures.

Exclusion criteria were self-reported cognitive impairments and self-reported psychiatric disorders, including a history of depression and schizophrenia. Multiple participation was not prohibited. However, the vast majority of participants took part in only one of the three studies. One single participant dropped out from study three (due to lack of interest).

The examiner provided no information regarding specific details about the aims of the study (i.e., examining ESE) to the participants in order to limit possible interferences with their performance. Rather than that, the participants received a general explanation of the aims of the study (i.e., examining limitations of cognitive flexibility) that did not mention ESE.

### 2.2. The Computerized Wisconsin Card-Sorting Task (cWCST)

Typical stimulus displays on cWCST trials are illustrated in Figure 1. The Introduction contains a detailed explanation of the basic task requirements. Here, we wish to add that we utilized only those 24 distinct Wisconsin target cards, which solely shared one feature with any of the key cards (those 24 target cards were selected from the original 64 Wisconsin target cards for the first time by [17]). The major advantage of selecting those 24 target cards is that the applied rule becomes unambiguously identifiable on any trial. For example, the target card depicting ‘four green crosses’ shares the feature ‘four’ with the outside-right key card, the feature ‘green’ with the inside-left key card, and the feature ‘crosses’ with the inside-right key card (see Figure 1A). Thus, the selected key card unambiguously informs about the rule that guided the selection on each trial. On the exemplified trial, selecting the outside-right key card allows to infer that the NUMBER rule was applied; selecting the inside-left key card allows to infer that the COLOR rule was applied; selecting the inside-right key card allows to infer that the SHAPE rule was applied on the trial.

Participants selected key cards by pressing corresponding response buttons. For further details, see the Method sections of Studies 1 to 3. The stimulus displays remained on screen until a response button was pressed. A blank screen followed the button press for a period of 800 ms before the visual feedback was presented, i.e., either the word ‘REPEAT’ or the word ‘SWITCH’ (worded in the German language). The REPEAT feedback indicated that the correct key card had been selected on the current trial, and hence, that the applied rule should be repeated on the next trial. The SWITCH feedback indicated that an incorrect key card had been selected on the current trial, and that switching away from the applied rule was requested on the following post-switch trial. Feedback stimuli remained on screen for 400 ms, and blank screens were presented for 800 ms following their offset.

Prior to data collection, all participants received verbal information about the three viable sorting rules, i.e., COLOR, SHAPE, and NUMBER rules were explicitly mentioned. They also received the verbal instruction that the to-be prioritized rule should be switched ‘once in a while’. Actually, the to-be prioritized rules switched unpredictably after two or more applications of the correct rule on successive trials [18]. All studies were programmed by means of Presentation software (Neurobehavioral Systems, Berkeley, CA, USA, https://www.neurobs.com, accessed on 7 January 2019).

### 2.3. Statistical Analyses

All trials on which participants committed odd errors as well as all trials that followed odd errors were excluded from statistical analyses. Selecting the key cards that did not share any feature with target cards are considered as odd errors. On the exemplified trial, selecting the outside-left key card would constitute an odd error because the target card (‘four green crosses’) and this key card (‘one red triangle) do not share any feature (see Figure 1A). The appearance of odd errors may be considered as being due to momentary lapses of attention. The frequency of odd errors was fortuitously extremely low, indicating that participants performed the Wisconsin card-sorting task in a very attentive manner. Exclusion of post-error trials is standard in cognitive psychology. We also excluded all trials in which identical target cards were presented on successive trials. The repetition of identical target cards occurred at a chance probability of *p* = 1/24 of all trials.

Conditional error probabilities served as dependent behavioral measures. Conditional error probabilities refer to the number of errors committed divided by the number of trials on which this error was possible. One conditional probability quantified repetitive PE. It quantifies the number of PE on repetitive trials divided by the number of all trials on which repetitive PE were possible (see Figure 1A). The other conditional probability quantified non-repetitive PE. It quantifies the number of PE on non-repetitive trials divided by the number of all trials on which non-repetitive PE were possible (see Figure 1A). ESE refers to the finding of reduced conditional probability of repetitive PE compared to conditional probability of non-repetitive PE.

Frequentist null-hypothesis testing uses *p*-values to quantify the probability of sample information given that the null hypothesis represents a valid model of the states of nature. Bayesian hypothesis testing provides a mechanism for combining a prior probability distribution for the states of nature with sample information to provide a revised (posterior) probability distribution about the states of nature. Bayes factors are ratios that are used to quantify the posterior support for one model over another model. Bayesian statistics may be considered superior to frequentist statistics from an epistemological perspective; however, these topics and their vast methodological consequences are of course hotly debated. We opted for Bayesian statistics and used the software package JASP (The JASP Team, Amsterdam, NL, The Netherlands, https://jasp-stats.org/, accessed on 6 January 2020) for Bayesian hypothesis testing. JASP is accompanied by an easily understandable introduction into Bayesian hypothesis testing and Bayes factors, which replace frequentist *p*-values (https://jasp-stats.org/wp-content/uploads/2018/07/USP_workshop.pdf, accessed on 6 January 2020).

We utilized Bayesian paired-samples *t*-tests to quantify evidence for the presence of ESE. Specifically, we computed Bayes factors with regard to the hypothesis that the conditional error probability of repetitive PE is reduced compared to the conditional error probability of non-repetitive PE (i.e., *BF_repetition<alteration_*). For interpretation of evidential strengths, we followed the convention: *BF_repetition<alteration_* > 3 represents substantial evidence for the presence of ESE, *BF_repetition<alteration_* > 10 represents strong evidence for the presence of ESE, and *BF_repetition<alteration_* > 100 represents extreme evidence for the presence of ESE. *BF_repetition<alteration_* < 1/3 represents substantial evidence against the presence of ESE, *BF_repetition<alteration_* < 1/10 represents strong evidence against the presence of ESE, and *BF_repetition<alteration_* < 1/100 represents extreme evidence against the presence of ESE.

Bayesian data analysis was performed using JASP Version 0.11.1 (The JASP Team, Amsterdam, NL, The Netherlands, https://jasp-stats.org/, accessed on 6 January 2020). We used JASP’s default settings for Bayesian paired samples *t*-tests, including Cauchy prior distributions with a width of 0.707.

## 3. Study 1

Study 1 examined the efficiency of one of the most parsimonious accounts of ESE. Specifically, cognitive inhibition accounts are often propelled forward in the context of executive functioning [19,20,21,22,23,24,25,26,27]. Inhibitory accounts of ESE converge in assuming inhibitory processing of the key cards that were selected or of the responses that were executed on trial *t* − 1. Inhibitory processing occurs putatively consequent to the negative feedback stimuli that were received on these switch trials (irrespective of feedback wording, i.e., ‘INCORRECT’ or ‘SWITCH’). The idea of inhibitory processing of the selected key cards/executed responses provides a very simple account of ESE. The commitment of repetitive PE—involving repetitions of the key cards and of the responses that received inhibitory processing—should be hindered to some degree compared to committing non-repetitive PE, which involve altered key cards and responses (see Figure 1A). Cognitive inhibition is hence suitable to explain ESE because our ESE studies revealed that repetitive PE were actually reduced compared to non-repetitive PE [9,10] (see Figure 1B).

Inhibitory accounts of ESE differ with regard to assumptions about which pieces of information receive inhibitory processing. First, inhibition may relate to the selected key card on trial *t* − 1. More specifically, inhibitory processing may target the key-card feature that objectified the to-be-prioritized rule on these key cards. In the example of Figure 1A, the ‘greenness’ of the stimuli depicted on the selected key card objectify the COLOR rule. Repetition of that feature on trial *t* may retrieve the inhibitory processing related to this feature such that repetitive PE become less prevalent than non-repetitive PE. In similar vein, inhibitory processing may target the response that served to select suitable key cards. Spatial response codes may be of particular relevance here. In the example of Figure 1A, selecting the key card that is suitable for the COLOR rule requires executing an ‘inside-left’ response on trial *t* − 1. Repetition of the requested spatial response on trial *t* may retrieve the inhibitory processing related to this response such that repetitive PE become less prevalent than non-repetitive PE. Inspection of Figure 1A reveals that the standard ESE involves conjoint repetitions of relevant features and of executed responses. We therefore refer to the standard ESE as the conjunctive ESE throughout the rest of this paper.

Inhibitory accounts of ESE conjecture that inhibitory processing related to relevant features or to executed responses are each sufficient to elicit ESE. However, the standard ESE design does not allow the disentangling effects of feature-related and of response-related inhibition on ESE. The design of Study 1—illustrated in Figure 2 (top panels)—circumvents this limitation. As can be seen from inspection of Figure 2, we simply rescheduled the allocation of the four key cards to spatial positions on each trial randomly. The left column of Figure 2 shows trials on which the previously selected key cards accidentally retained their spatial positions. These occasions allow testing the replicability of conjunctive ESE, because repetitive PE involve repetition of relevant features and of executed responses on these occasions. The remaining columns of Figure 2 show trials on which the previously selected key cards altered their spatial positions. All PE trials in the central column involve response alteration. Some of these trials involve repetition (versus alteration) of the relevant feature such that the comparison between the prevalence of PE on these trials yields insight into the efficiency of feature-related inhibition. These occasions allow testing the effects of feature-related inhibition on ESE, because repetitive PE involve repetition of relevant features, but not of executed responses, on these occasions. All PE trials in the right column involve feature alteration. Some of these trials involve repetition (versus alteration) of the executed response such that the comparison between the prevalence of PE on these trials yields insight into the efficiency of response-related inhibition. These occasions allow testing the effects of response-related inhibition on ESE, because repetitive PE involve repetition of executed responses, but not of relevant features, on these occasions. We refer to either feature-based or response-based ESE as disjunctive ESE throughout the rest of this paper.

The design of Study 1 allows for examining not only conjunctive ESE (as in standard designs), but its simple trial-by-trial manipulation of spatial key-card positions renders it possible to examine the effects of feature-related and of response-related inhibitory processing on ESE in isolation. In essence, inhibitory accounts of ESE predict the existence of disjunctive ESE, be that feature-based ESE or response-based ESE.

### 3.1. Methods

#### 3.1.1. Participants

A total of 40 undergraduate students (32 female) participated in Study 1. They received a payment of EUR 10 per hour. The mean age of participants was 23.83 years (*SD* = 4.36 years). All participants had normal or corrected-to-normal vision.

#### 3.1.2. cWCST Manipulation

We randomly rescheduled the spatial position of each key card on each trial. That is, key cards appeared randomly at any spatial position (i.e., outside-left, inside-left, inside-right, or outside-right) on a particular trial. Figure 2 (top panels) provides examples of the trial-by-trial manipulation of the spatial positioning of key cards. Note that any key card could either retain or alter its spatial position compared to its position on the previous trial.

Participants selected key cards by pressing response buttons that were spatially mapped to key-card positions. That is, key cards presented at the outside-left, the inside-left, the inside-right, and the outside-right position were selected by pressing the outside-left, the inside-left, the inside-right, or the outside-right response button, respectively. Participants pressed the outside-left, the inside-left, the inside-right, and the outside-right response button using the left middle finger, the left index finger, the right index finger, or the right middle finger, respectively. We collected button presses with a Cedrus response pad RB 830 (Cedrus, San Pedro, CA, USA, https://cedrus.com, accessed on 7 January 2019).

#### 3.1.3. Procedure

Study 1 comprised five blocks of trials. Each block included 40 rule switches. Prior to the first block of trials, participants completed a training block that included five rule switches.

#### 3.1.4. Analysis

We analyzed conditional PE probabilities for each individual. First, we analyzed those trials on which the previously selected key card retained its spatial position. We focused on comparing repetitive and non-repetitive PE, as illustrated in the left column of Figure 2. Second, we analyzed trials on which the previously selected key card altered its spatial position. The central column of Figure 2 illustrates one contrast of interest: Committing PE imply response alteration on all these occasions. Repetitive PE imply repetition of the relevant feature, and non-repetitive PE imply alteration of the relevant feature. The right column of Figure 2 illustrates the other contrast of interest: Committing PE imply feature alteration on all these occasions. Repetitive PE imply response repetition, and non-repetitive PE imply response alteration.

Table 1 presents the frequency (average number of trials and variability) with which these various types of PE could possibly occur.

### 3.2. Results

The results from Study 1 are presented in Figure 2 (bottom panels). On trials on which previously selected key cards retained their spatial positions, we found substantial evidence for conjunctive ESE (*BF_repetition<alteration_* = 8.282). Repetitive PE were reduced (*M* = 0.051 (conditional probability); *SE* = 0.010) compared to non-repetitive PE (*M* = 0.081 (conditional probability); *SE* = 0.009; see left column of Figure 2). Thus, we found substantial evidence for the presence of conjunctive ESE, replicating our previous results that were obtained from larger samples [9,10].

Next, we analyzed trials in which previously selected key cards altered their spatial positions. Comparing PE on trials involving repetition (versus alteration) of the relevant feature in the presence of response alteration yielded substantial evidence against the hypotheses of feature-based disjunctive ESE (*BF_repetition<alteration_* = 0.261; central column of Figure 2). Comparing PE on trials involving repetition (versus alteration) of the executed response in the presence of the feature alteration yielded substantial evidence against the hypotheses of response-based disjunctive ESE (*BF_repetition<alteration_ =* 0.187; right column of Figure 2).

### 3.3. Discussion

As outlined in the Introduction to Study 1, there are two plausible alternatives how cognitive inhibition [19,20,21,22,23,24,25,26,27] may account for ESE: disjunctive ESE are expected according to both qualifications of inhibition, be that feature-based ESE as a result of feature-related inhibition, or response-based ESE as a result of response-related inhibition. The results clearly show that no evidence was found to support this essential prediction from inhibitory accounts of ESE. The complete lack of evidence for disjunctive ESE stands in sharp contrast to the successful replication of conjunctive ESE in Study 1. One can conclude that cognitive inhibition is ruled out as an explanatory concept for ESE, as long as one considers inhibition is as an elemental (feature- or response-related) process. The idea that inhibitory processes could be tagged to relevant key-card features *or* to executed responses does not clearly provide a suitable explanation for the presence of conjunctive ESE in the absence of disjunctive ESE.

Cognitive inhibition may, however, serve as an explanatory principle of the pattern of ESE that we report here if one conceptualizes it as occurring strictly in the service of conjunctive control [28]. Studying the role of conjunctive internal representations has a long history in theories of associative learning [29]. In fact, the formation of associations between stimuli (as in the case of classical (also Pavlovian) conditioning) or between stimuli and responses (as in the case of instrumental (also operant) conditioning) can be conceived in terms of regularities how conjunctive internal representations exert control over behavior [30]. Study 2 was conducted to gain a more complete understanding of the exact nature of the conjunctive relations within internal representations that gave rise to conjunctive ESE in the absence of disjunctive ESE.

## 4. Study 2

Study 2 examined the efficiency of associative accounts of conjunctive ESE. Theories of associative learning can be divided into those theories that attribute behavioral changes to the formation of associations between stimuli (as in the case of classical (also Pavlovian) conditioning; [31]) and those theories that attribute behavioral changes to the formation of associations between stimuli and responses (as in the case of instrumental (also operant) conditioning; [15]). The following paragraphs apply these broad associative themes to Wisconsin card-sorting tasks for a more complete understanding of conjunctive ESE.

The commitment of repetitive PE, but not of non-repetitive PE, involves reenactment of the previously selected key-card features (see Figure 1A). In addition to that affordance, conjunctive ESE imply that spatial key-card positions are retained: recall that Study 1 showed that the stability of this spatial structure is a necessary condition for the occurrence of ESE. One possibility is that associative memories for stimulus (feature)–stimulus (position) associations may be relevant for conjunctive ESE. Thus, the formation of stimulus–stimulus (S-S) associations, in particular associations between key-card features and key-card positions, may contribute to conjunctive ESE. In the example depicted in Figure 1A, the S-S association between ‘green’ key card (feature) and ‘inside-left’ key card (position; formed on trial *t* − 1) would be reenacted on trial *t* in the case of repetitive PE, but not in the case of non-repetitive PE.

Readers may have noticed that spatial positions of key cards are confounded with spatial codes of responses. As an example, ‘inside-left’ key cards correspond to ‘inside-left’–responses. Therefore, an alternative possibility is that associative memories for stimulus (feature)–response (position) associations may be relevant for conjunctive ESE. Thus, the formation of stimulus–response (S-R) associations, in particular associations between key-card features and spatial response codes, may contribute to conjunctive ESE. In the example depicted in Figure 1A, the S-R association between ‘green’ key card (feature) and ‘inside-left’–response (formed on trial *t* − 1) would be reenacted on trial *t* in case of repetitive PE, but not in case of non-repetitive PE.

The essential idea behind Study 2 was to un-break the named confounding. The desired de-confounding was simply achieved through manipulating S-R mapping [32], i.e., through the manipulation of effective S-R mappings via a random trial-by-trial schedule, as illustrated in Figure 3. As an example, selecting ‘inside-left’ key cards requested to press ‘button 2’ according to one S-R mapping and to press ‘button 3’ according to the other S-R mapping. An additional mapping cue on each trial indexed the currently effective S-R mapping (see Figure 3). It was therefore possible to separate trials on which the effective S-R mappings were retained (constant S-R mappings) from trials on which the effective S-R mappings were altered (varied S-R mappings).

The design of Study 2 allows for de-confounding S-S and S-R associative accounts of conjunctive ESE. The conditions of constant S-R mappings (left column of Figure 3) correspond to standard designs, thereby providing another opportunity for testing the replicability of conjunctive ESE. Varied S-R mappings (right column of Figure 3) actually invited de-confounding S-S and S-R associative accounts of conjunctive ESE. Learning S-S associative memories (on trials *t* − 1 and reenacting them on trials *t*) predicts the presence of conjunctive ESE on these occasions because the selected key card reappears at its previously held spatial position. Learning S-R associative memories (on trials *t* − 1 and reenacting them on trials *t*) predicts the absence of conjunctive ESE on these occasions because a different response to the previously executed response is requested by the altered S-R mapping rule.

### 4.1. Methods

#### 4.1.1. Participants

A total of 40 undergraduate students (28 female) participated in Study 2. They received a payment of EUR 10 per hour. The mean age of participants was 23.75 years (*SD* = 3.60 years). All participants had normal or corrected-to-normal vision.

#### 4.1.2. cWCST Manipulation

Key cards were presented horizontally at constant positions (i.e., across all trials, the key card depicting one red triangle, two green stars, three yellow crosses, and four blue circles appeared at the outside-left, the inside-left, the inside-right, and the outside-right position, respectively). Responses were aligned vertically. We manipulated the mapping between key-card positions and corresponding responses from trial to trial. There were two different mappings between key-card positions and corresponding responses.

According to the left-right to up-down mapping, participants selected the outside-left key card by pressing the topmost button (1), the inside-left key card by pressing the meso-upper button (2), the inside-right key card by pressing the meso-lower button (3), and the outside-right key card by pressing the lowermost button (4). For an illustration of the left-right to up-down mapping, see trial *t* on the left column of Figure 3. According to the left-right to down-up mapping, participants selected the outside-left key card by pressing the lowermost button (4), the inside-left key card by pressing the meso-lower button (3), the inside-right key card by pressing the meso-upper button (2), and the outside-right key card by pressing the topmost button (1). For an illustration of the left-right to down-up mapping, see trial *t* on the right column of Figure 3.

Response buttons were aligned vertically in order to avoid that one S-R mapping possessed a higher stimulus–response compatibility than the other (i.e., in order to avoid comparing the left-right to left-right mapping with the left-right to right-left mapping). We collected button presses with a vertically arranged keypad. Participants utilized their little finger, ring finger, middle finger, and index finger of the dominant hand for pressing the topmost button (1), the meso-upper button (2), the meso-lower button (3), and the lowermost button (4), respectively.

Auditory mapping cues were presented (duration = 100 ms) 200 ms prior to the onset of target displays. 600 Hz sounds requested applying the left-right to up-down mapping. Sounds of 350 Hz requested applying the left-right to down-up mapping. The probability of mapping switches amounted to 75% from trial to trial. The relatively high probability of mapping switches was chosen to ensure that sufficient numbers of mapping switches remained available for data analysis.

#### 4.1.3. Procedure

Study 2 comprised five blocks of trials. Each block included 40 rule switches. Prior to the first block of trials, participants completed three training blocks. The first training block included three rule switches, and participants applied the left-right to up-down mapping throughout the whole block. The second training block included three rule switches, and participants applied the left-right to down-up mapping throughout the whole block. The final training block included five rule switches. Only on this training block, S-R mappings altered between the left-right to up-down mapping and the left-right to down-up mapping as indicated by the trial-specific mapping cues.

#### 4.1.4. Analysis

We analyzed conditional PE probabilities for each individual. First, we analyzed those trials on which the S-R mapping was retained. We focused on comparing repetitive and non-repetitive PE, as illustrated in the left column of Figure 3. Second, we analyzed trials on which the S-R mapping switched. The right column of Figure 3 illustrates the contrast of interest: Committing PE implies response alteration on all these occasions: Repetitive PE imply repetition of the relevant feature, and non-repetitive PE imply alteration of the relevant feature.

There is one additional complexity involved in this study, which is that S-R mappings may remain maintained even though mapping cues signaled that S-R mappings should be altered. Conclusions from behavior on such trials may be misleading. The right columns of Figure 3 illustrate the issue. On these occasions (trials *t*), erroneously retaining the previously valid ‘up-down’ S-R mapping plus committing an odd error (see above for a definition of odd errors) under this S-R mapping equals a *PE* under the actually valid ‘down-up’ S-R mapping. We solely analyzed data from those trials where an observed PE—assuming that the correct S-R mapping was applied—would equal an odd error in case that the S-R mapping remained maintained even though the current mapping cue signaled that the S-R mapping should be altered. Observed PE on these trials are likely indicating actual rule perseveration under the application of the correct S-R mapping. The alternative possibility, i.e., untruly S-R mapping maintenance plus commitment of an odd error, seems very unlikely due to the extreme rarity of odd errors. Due to the exclusion of trials that did not meet these criteria, the number of analyzed trials with altered S-R mappings was lower than the number of analyzed trials with retained S-R mappings (see Table 2) despite the fact that the former type of trials (*p* = 0.75) actually occured more frequently than the latter type of trials (*p* = 0.25). Table 2 presents the frequency (average number of trials and variability) with which these various types of trials were analyzed.

### 4.2. Results

The results of Study 2 are presented in Figure 3 (bottom panels). On the trials with retained S-R mappings, we found substantial evidence (*BF_repetition<alteration_* = 3.059) for reduced repetitive PE (*M* = 0.055; *SE* = 0.011 (conditional probability)) compared to non-repetitive PE (*M* = 0.079; *SE* = 0.011 (conditional probability); see left column of Figure 3). Thus, we found substantial evidence for the presence of conjunctive ESE.

On the trials with altered S-R mappings, we found substantial evidence (*BF_repetition<alteration_* = 0.141) against the hypothesis of reduced repetitive PE compared to non-repetitive PE. Please keep in mind that on the trials with altered S-R mappings, the term ‘repetitive’ refers to feature repetition and response alteration due to the altered S-R mappings on those trials (see right column of Figure 3).

### 4.3. Discussion

As outlined in the Introduction to Study 2, there are two plausible alternatives as to how associative learning may account for conjunctive ESE. If S-S associative memories provide an efficient explanatory concept for ESE, they would emerge under constant and varied S-R mappings. In contrast, if S-R associative memories provide an efficient explanatory concept for ESE, they would emerge under constant S-R mappings, but *not* under varied S-R mappings. The results clearly favor the latter possibility. Conjunctive ESE were solely found on standard occasions (constant S-R mappings), further supporting the replicability of the phenomenon. The complete lack of evidence for conjunctive ESE on trials that comprised varied S-R mappings suggests that S-R—rather than S-S—learning provides a suitable explanation for conjunctive ESE.

How exactly do instrumental S-R associations explain conjunctive ESE on Wisconsin card-sorting tasks? To begin with, the traditional instrumental theory assumes that S-R associative memories are formed on trials *t* − 1. In more detail, these instrumental memories may comprise S (feature)-R (spatial code) associations. In the example that is repeatedly made in all of our Figures, the instrumental memory on trial *t* − 1 comprises the association between S (‘green’) and R (‘inside-left’). Reenactment on trial *t* under exactly identical conditions (in terms of conjunctions: S (feature = ‘green’) and R (spatial code = ‘inside-left’)) triggers the retrieval of this instrumental associative memory.

Readers may have noticed that the instrumental explanation requires the idea of inhibition: the retrieval of instrumental memories should interfere with, rather than facilitate, the behavioral expression of recently memorized instrumental associations. This inhibitory assumption is necessary because conjunctive ESE reflect the fact that repetitive PE are reduced compared to non-repetitive PE. At first glance, the inhibitory assumption seems to be at odds with what is widely known about instrumental learning. As explained above, instrumental learning is often conceived as the acquisition of habits through the formation of associations between stimuli and responses (outcomes are considered as mere catalyzers). Once acquired, habits, i.e., acquired S-R associations, may gain behavioral control in the absence of additional deliberate control [14,15,16]. Habitual behavioral control actually predicts reduced non-repetitive PE compared to repetitive PE, and this prediction was clearly disconfirmed in all our hitherto existing ESE studies.

However, dual-process models of instrumental learning conceive two separable levels of instrumental memories [14,15,16]. Specifically, the *goal-directed* level of instrumental learning is considered a remaining contingent upon outcome monitoring: whereas habitual instrumental memories hold bipartite (S-R) associations, goal-directed instrumental memories comprise tripartite (S-R-outcome (O)) associations. Tripartite goal-directed instrumental memories are well capable to explain conjunctive ESE on Wisconsin card-sorting tasks because—further pursuing the above example—the tripartite goal-directed instrumental memory on trial *t* − 1 comprises the association between S (‘green’) and R (‘inside-left’) and O (‘switch’). Reenactment on trial *t* under exactly identical conditions (in terms of conjunctions: S (feature = ‘green’) and R (spatial code = ‘inside-left’)) triggers the retrieval of this tripartite goal-directed instrumental association, including its O (‘switch’) element, thereby hindering the commitment of repetitive PE.

We conclude from Study 2 that goal-directed instrumental learning provides an efficient explanation for conjunctive ESE. Conjunctive ESE on Wisconsin card-sorting tasks seem to emerge from shifts in the balance between habitual and goal-directed instrumental learning. According to this account, repetitive PE are reduced compared to non-repetitive PE because the retrieval of tripartite (S-R-O) instrumental memories is more efficient on repetitive occasions, which involve the repetition of S-R conjunctions. Conjunctive ESE hence reflect more efficient goal-directed instrumental control over PE on repetitive occasions. These thoughts will be elaborated on in the Concluding Discussion, in which we present a novel theory of cognitive perseveration (i.e., the *g*oal-directed *i*nstrumental *c*ontrol GIC model). Interested readers may also take note of our computational studies, in which we analyzed the explanatory power of formalized dual-level learning models of PE and ESE on Wisconsin card-sorting tasks [10,13,33,34,35].

The instrumental theory of conjunctive ESE on Wisconsin card-sorting tasks draws attention to goal-directed instrumental control via the retrieval of tripartite instrumental memories [15]. Study 3 was conducted to gain a more complete understanding of the exact nature of the concept of ‘responses’, which constitute elements of these tripartite instrumental memories.

## 5. Study 3

The design of Study 3 allows for a more complete understanding of the nature of instrumental memories by clarifying the meaning of ‘R’. The acronym ‘R’ may represent a spatial response code at a ‘central’ (cognitive) level that does not yet specify the effectors that are required to achieve the desired response. As an alternative, ‘R’ may represent the effector-specific response that needs to be performed and that exists at a somewhat ‘lower’ (motor) level.

Study 3 manipulated effective response–effector (R-E) mappings [36] in a random trial-by-trial schedule, as illustrated in Figure 4. As an example, selecting ‘inside-left’ key cards requested to press ‘button 2’ with the right hand according to one R-E mapping and to press ‘button 2’ with the left hand according to the other S-R mapping. An additional mapping cue on each trial indexed the currently effective R-E mapping (see Figure 4). It was therefore possible to separate trials on which the effective R-E mappings were retained (constant R-E mappings) from trials on which the effective R-E mappings were altered (varied R-E mappings).

The conditions of constant R-E mappings (left column of Figure 4) correspond to standard designs, once again providing another opportunity for testing the replicability of conjunctive ESE. Varied R-E mappings (right column of Figure 4) actually invited separating cognitive and motor specifications of ‘R’. The cognitive understanding of ‘R’ predicts the presence of conjunctive ESE on varied R-E mapping trials because spatial codes could be repeated (irrespective of the utilized effectors). The motor understanding of ‘R’ predicts the absence of conjunctive ESE on these occasions exactly because they do request effector switches.

### 5.1. Methods

#### 5.1.1. Participants

A total of 41 undergraduate students (27 female) participated in Study 3. They received a payment of EUR 10 per hour. The mean age of participants was 23.76 years (*SD* = 3.67 years). The data from one participant had to be deleted upon this participant’s request. The mean age of the final sample (*N* = 40; 27 female) was 23.70 years (*SD* = 3.70 years). All participants had normal or corrected-to-normal vision.

#### 5.1.2. cWCST Manipulation

The responding hands varied randomly on a trial-by-trial basis such that participants pressed response buttons either with their right hand or with their left hand. Button presses were collected via two numeric keypads that were aligned horizontally. They were superimposed in such a way that the lower keypad was mounted opposite of the upper keypad. Each keypad was utilized via four fingers of one hand. Hands had to be rotated face to face such that identical fingers of both hands would overlap each other in the absence of keypads (readers may think of ‘praying’ hands for an appropriate imagination of this hand position). Thus, participants utilized their index finger (1), middle finger (2), ring finger (3), and little finger (4) of either hand for selecting specific key cards, as detailed below.

The assignment of hands to keypads was counterbalanced across participants. Fifty percent of the participants operated the upper keypad with their right hand and the lower keypad with their left hand; this arrangement would be achieved by rotating ‘praying’ hands counterclockwise. These participants utilized their index fingers, middle fingers, ring fingers, and little fingers of both hands for selecting outside-left, inside-left, inside-right, and outside-right key cards, respectively. Fifty percent of the participants operated the upper keypad with their left hand and the lower keypad with their right hand; this arrangement would be achieved by rotating ‘praying’ hands clockwise. These participants utilized their little fingers, ring fingers, middle fingers, and index fingers of both hands for selecting outside-left, inside-left, inside-right, and outside-right key cards, respectively.

Auditory hand cues were presented (duration = 100 ms) 200 ms prior to the onset of target displays. Sound levels of 400 Hz were presented either via the left or via the right earbud. Hand cues presented on the left earbud requested left-hand responses, and hand cues presented on the right earbud requested right-hand responses. Repeating hands/switching hands was requested with equal probabilities.

#### 5.1.3. Procedure

Study 3 comprised four blocks of trials. Each block included 40 rule switches. Prior to the first block of trials, participants completed three training blocks. The first training block included three rule switches, and participants pressed response buttons with the upper hand only. The second training block included three rule switches, and participants pressed response buttons with the lower hand only. The final training block included five rule switches. It was only on this training block where the responding hands varied from trial to trial as indicated by the hand cues.

#### 5.1.4. Analysis

We analyzed conditional PE probabilities for each individual. First, we analyzed those trials on which the R-E mapping was retained (hand repetitions). We focused on comparing repetitive and non-repetitive PE, as illustrated in the left column of Figure 4. Second, we analyzed trials on which the R-E mapping switched (hand alterations). The right column of Figure 4 illustrates the contrast of interest: despite altered effectors, it was possible to compare repetitive and non-repetitive PE on these trials.

Table 3 presents the frequency (average number of trials and variability) with which these various types of PE could possibly occur. Notice that we excluded all trials on which participants pressed a response button with the wrong hand (5.73% of all trials).

### 5.2. Results

The results of Study 3 are presented in Figure 4 (bottom panels). On the trials with retained R-E mappings (i.e., identical hands responding), we found substantial evidence (*BF_repetition<alteration_* = 4.718) for reduced repetitive PE (*M* = 0.061 (conditional probability); *SE* = 0.010) compared to non-repetitive PE (*M* = 0.082 (conditional probability); *SE* = 0.011); see left column of Figure 4. Thus, we found substantial evidence for the presence of a conjunctive ESE when the responding hand remained the same across trials.

On trials with altered R-E mapping (i.e., different hands responding), we found extreme evidence (*BF_repetition<alteration_* = 102.137) for reduced repetitive PE (*M* = 0.057 (conditional probability); *SE* = 0.011) compared to non-repetitive PE (*M* = 0.087 (conditional probability); *SE* = 0.012); see right column of Figure 4. Thus, we found extreme evidence for the presence of a conjunctive ESE even though the responding hand altered across trials.

### 5.3. Discussion

As outlined in the Introduction to Study 3, there are two plausible conceptualizations of the responses (‘R’) that form part of instrumental memories on Wisconsin card-sorting tasks. ‘R’ may represent an effector-free cognitive construct, or alternatively, it may represent an effector-specific motor construct. The results clearly favor the former possibility. Conjunctive ESE were found on standard occasions (constant R-E mappings), once again supporting the replicability of the phenomenon. The clear evidence for conjunctive ESE on trials that comprised varied R-E mappings suggests that ‘R’—entering instrumental associative memories—should be conceived as being cognitive in nature. ‘R’ on Wisconsin card-sorting tasks may putatively be specified in terms of rather abstract spatial codes (such as ‘inside-left’ and the like).

The cognitive nature of ‘R’ in instrumental associative memories on Wisconsin card-sorting tasks suggests that instrumental associative learning occurs at relatively high cognitive levels. The Concluding Discussion provides some additional thoughts about this and other findings that we obtained from our studies.

## 6. Summary and Concluding Discussion

Kopp et al. [9] first demonstrated that repetitive PE are reduced compared to non-repetitive PE (i.e., the presence of conjunctive ESE) on a paper-and-pencil version of Wisconsin card-sorting tasks in a clinical setting. Steinke et al. [10] replicated this finding in a quite different context, i.e., on a computerized version of Wisconsin card-sorting tasks in a non-clinical setting. All three studies that were presented here yielded additional replications of conjunctive ESE on standard conditions of computerized versions of Wisconsin card-sorting tasks in non-clinical settings. The first conclusion is that conjunctive ESE are a well-replicable phenomenon on Wisconsin card-sorting tasks. This conclusion can be drawn with relatively high confidence in the face of its multiple replications.

The three studies that were presented here also yielded first insights into which theoretical perspectives are best suited for explaining the phenomenon. Study 1 revealed that while conjunctive ESE constitute a well-replicable phenomenon, no evidence for disjunctive ESE emerged. The dissociation between conjunctive and disjunctive ESE pointed to the direction of associative accounts for conjunctive ESE. Study 2 clearly favored instrumental (S-R) over classical (S-S) associative learning if one assumes the formation of tripartite, goal-directed instrumental memories (i.e., S-R-O associations). Viewed from this perspective, conjunctive ESE provide insight into mechanisms of goal-directed instrumental control, which is more efficient on repetitive than on non-repetitive occasions for cognitive perseveration. Thus, goal-directed instrumental control, which involves the retrieval of tripartite, goal-directed (i.e., S-R-O) memories via bipartite retrieval cues (i.e., repeated S-R conjunctions), may be best suited for explaining the phenomenon of conjunctive ESE (as detailed below). Study 3 suggested that the formation of tripartite, goal-directed instrumental memories, despite being associative in nature, occurs in the service of high-level cognition: responses seem to be encoded at an abstract, putatively spatial, level that does not yet specify its effectors.

Conjunctive ESE cannot be reconciled with existing neuropsychological theories. We already discussed that cognitive inhibition per se [19,20,21,22,23,24,25,26,27] cannot explain conjunctive ESE in the absence of disjunctive ESE. Other neuropsychological theories are plagued with similar shortcomings. For example, one of the most popular theories in the field is the ‘supervisory attentional system’ (SAS) theory [37,38,39]. The SAS theory postulates that behavior is controlled by ‘contention scheduling’ on routine occasions (contention scheduling shares many characteristics with habitual instrumental control). Contributions from the SAS are primarily requested under conditions of novelty. The SAS and similar dual-process theories that distinguish between automatic and controlled processing may explain why switch trials pose additional challenges to behavioral control compared to non-switch trials. None of these dual-process theories, however, explains conjunctive ESE. If anything, they predict reverse conjunctive ESE because repetitive PE (in a sense, occurring on ‘routine’ situations) should be subject to automatic processing (or contention scheduling) with the effect of enhanced error proneness, whereas non-repetitive PE (in a sense, occurring on ‘novel’ situations) should be subject to controlled processing (achieved through the SAS and the like) with the effect of reduced error proneness. As a consequence, the SAS and similar dual-process theories predict reverse conjunctive ESE.

The triarchic theory of learning [40,41] offers at first sight a somewhat more promising account of conjunctive ESE. Following the initial formation of strategies that support task execution, learners transiently activate a controlled-execution system (this happens presumably on post-switch trials), which guides action selection via reinforcement signals (e.g., outcomes). Later, the gradual strengthening of key S-R associations that underlie task execution allows cognitive-control resources to slowly disengage as task performance becomes practiced and, eventually, automatic. However, as with dual-process models, the triarchic theory of learning cannot account for reduced repetitive PE compared to non-repetitive PE. This theoretical shortcoming is due to the fact that repeated task execution is thought to favor more rapidly evolving automaticity, thereby enhancing error proneness during transitions between controlled and automatic behavior.

The data that we presented in these three studies therefore call for a novel theory of cognitive perseveration. Figure 5 provides a flow-chart illustration of the novel theory, which we refer to as the *g*oal-directed *i*nstrumental *c*ontrol (GIC) model of cognitive perseveration. On switch trials, GIC comes into play through a feedback-based route and—eventually– through the repetition of S-R conjunctions on post-switch trials. The top panel shows a repetitive post-switch trial *t*, cognizable by the repetition of the identical S-R conjunction on *t* − 1 and on *t*. On repetitive trials, the repetition of identical S-R conjunctions activates GIC via the retrieval of tripartite S-R-O memories that were acquired on most recent switch trials. Hence, feedback-based and retrieval-based routes to GIC act together in inhibiting rule perseveration on repetitive post-switch trials, thereby strongly weakening the propensity of PE. The bottom panel shows a non-repetitive post-switch trial *t*, cognizable by the alteration of the S-R conjunction on *t* − 1 (S-R) and on *t* (S’-R’). On non-repetitive trials, only feedback-based GIC inhibits rule perseveration, thereby weakening the propensity of PE somewhat less efficiently. In a nutshell, repetitive PE are subject to additive (i.e., feedback-based plus retrieval-based) suppression from GIC, whereas non-repetitive PE are merely suppressed via feedback-based GIC. Repetitive PE ($p_{r}$) are therefore reduced compared to non-repetitive PE ($p_{n}$) across multiple trials, generating the empirically observable phenomenon of conjunctive ESE.

This novel theory differs from the above-mentioned dual-process models in substantial ways. Notice that the GIC model does not explain conjunctive ESE by distinguishing between qualitatively different processes (such as habitual versus goal-directed or automatic versus controlled and the like), though assuming dual processes may still have its merits (see below). The GIC model is a single-process model of conjunctive ESE. Specifically, the GIC model conjectures that the availability of dual routes to one single process—namely feedback-based and retrieval-based routes—modulates the behavioral efficiency of GIC in a quantitative manner. The GIC model relies on a single cognitive mechanism, for which it describes preconditions for modulating its behavioral efficiency. These preconditions, in turn, equal clearly operationalizable situations, which render the GIC model empirically testable.

The GIC model was a-posteriori induced from the data; this is why it is presented in the Concluding Discussion rather than in the Introduction of this paper. It nonetheless raises the claim that it may serve as an a-priori asserted theory in future studies because the GIC model is a bit more specific with regard to the antecedent and consequent conditions than existing theories. Although the GIC model is formulated in the relatively narrow context of Wisconsin card-sorting tasks, its scope of validity is the much broader theme of cognitive perseveration.

Another aspect of the GIC model deserves a short comment. Readers may have noticed that the meaning of ‘instrumental’ does not only concern relations between observable ‘stimuli’ and ‘responses’. With regard to stimuli (‘S’), one should keep in mind that the typical layout of the Wisconsin stimulus material is highly complex (see Figure 1). Perceiving ‘S’ implies identifying the rule-contingent, to-be-prioritized stimulus feature on any Wisconsin trial. Thus, disambiguation of target cards via the application of the correct rule is necessary, which is putatively achieved through mechanisms of selective attention [42]. When talking about ‘S’ in the context of ‘S-R’ or ‘S-R-O’ associative memories, we have this strongly pre-processed rule-contingent, to-be-prioritized feature in mind. In similar vein, the acronym ‘R’ describes the feature-contingent selection of corresponding key cards. As revealed by the results of Study 3, ‘R’ should not be understood in the effector space. Rather than that, ‘R’ describes the spatial code of the corresponding key card on any Wisconsin trial, which remains in the cognitive sphere. These two elements (‘S’ and ‘R’) of ‘instrumental’ associative memories request a cognitive extension of our understanding of ‘instrumental’ learning [43].

We also wish to call attention to the potential neuropsychological assessment of conjunctive ESE in individuals in future studies. Specifically, we suggest a normalized delta metric for its quantification, i.e.,
$$\Delta_{PE}=\frac{p_{n}-p_{r}}{p_{n}+p_{r}}$$
with $p_{n}$ quantifying the individual conditional probability of non-repetitive PE, and $p_{r}$ quantifying the individual conditional probability of repetitive PE (defined for all individuals who commit at least some PE, i.e., $p_{n}$ + $p_{r}\;\neq 0$). $\Delta_{PE}=1$ if $p_{n}=1$ and $p_{r}=0$ (max. ESE), $\Delta_{PE}=0$ if $p_{n}=p_{r}$ (no ESE), and $\Delta_{PE}=-1$ if $p_{n}=0$ and $p_{r}=1$ (max. reverse ESE).

Note that the numerator constitutes a difference (delta) measure that—on average—quantifies reduced repetitive PE compared to non-repetitive PE, i.e., group-level conjunctive ESE. Calculating the quotient (normalization) ensures that proportional (rather than absolute) PE reduction will be quantified. For example, assuming identical absolute PE reductions (i.e., identical numerators, 0.1 in the examples below), different measures of PE reduction will result in different overall PE propensities (denominators), i.e., $\Delta_{PE}=0.333$ if $p_{n}=0.2$ and $p_{r}=0.1$, but $\Delta_{PE}=0.067$ if $p_{n}=0.8$ and $p_{r}=0.7$.

A note of caution regarding the application of this normalized delta metric is required because it is a difference measure. Difference measures have the potential of providing highly valid indicators of those specific aspects of neuropsychological functioning that are under scrutiny. As an example, the suggested normalized delta metric has the potential to serve as a (relatively) pure measure of the individual strength of repetitive GIC. We already know from previous studies that overall PE propensities (i.e., denominators) are highly redundant measures of overall performance on Wisconsin card-sorting tasks [44]. Thus, the field needs more valid metrics for cognitive perseveration. However, the increased validity of difference measures usually comes at the cost of decreased reliability [45,46]. The consistency reliability of the suggested difference measure needs careful investigation [47,48,49].

Over and above these psychometric issues, one should avoid applying any metric thoughtlessly in neuropsychological assessment. Imagine a patient who perseverates one and the same rule on each single Wisconsin trial, with the consequence that $\Delta_{PE}=0$ because $p_{n}=p_{r}=1$. In these cases, the merit of dual-process models becomes evident: these patients show a complete preponderance of habitual (automatic) over goal-directed (controlled) behavior. In other words, very strong scenarios of imbalance between habitual over goal-directed behavior may actually be reflected to some degree in the strength of overall PE propensities (i.e., in the denominator).

## 7. Conclusions

Cognitive neuropsychology is currently characterized by poorly developed theories of executive functioning. Existing theories (e.g., see the examples above) lack sufficient specification of antecedent and consequent conditions such that they cannot be scrutinized scientifically in terms of falsification [50]. Relatedly, they often do not allow predicting the to-be-expected behavioral effects from operationalizable preconditions. Our novel theory of cognitive perseveration tries to overcome these shortcomings: starting from the well-replicable observation that PE on repetitive Wisconsin trials are reduced in comparison to non-repetitive PE, we induced the GIC model of cognitive perseveration from the reported data.

We have known since the pioneering work by Milner [4] and Luria [12] that cognitive perseveration should be considered to be a cardinal symptom following extensive prefrontal lesions. Surprisingly, however, there is not much systematic, or even theory-driven, research into functional and structural mechanisms of cognitive perseveration. We hope that the discovery of ESE and the development of the GIC model will stimulate neuropsychological research to improve our understanding of corresponding brain function–structure associations, which remains the ultimate goal of experimental neuropsychology.

## Figures and Tables

**Figure 1 brainsci-13-00919-f001:**
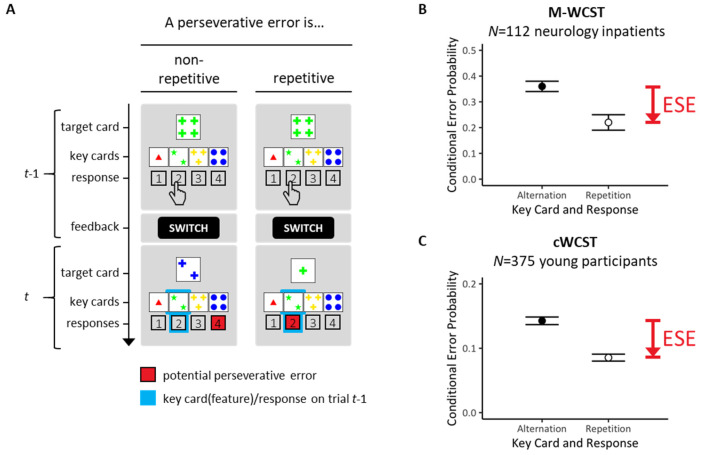
An outline of error-suppression effects (ESE). Figure 1A shows the typical layout of the Wisconsin stimulus material. All Wisconsin stimulus cards can be described in terms of a three-dimensional rule space (with the rules COLOR/SHAPE/NUMBER of the depicted items). Each rule is instantiated by four distinct features (with COLOR features: red, green, yellow, blue/SHAPE features: triangle, star, cross, circle/NUMBER features: #1, #2, #3, #4). On each trial, a response is requested that assigns the current target card to one of four simultaneously presented key cards (i.e., outside-left key card: ‘1 red triangle’, inside-left key card: ‘2 green stars’, inside-right key card: ‘3 yellow crosses’, outside-right key card: ‘4 blue circles’). The task is selecting—on each trial—the key card that shares the feature with the to-be-prioritized rule-contingent target feature (be that the COLOR, SHAPE or NUMBER feature). This is exemplified on trial *t* − 1 of Figure 1A by selecting the inside-left key card ‘2 green stars’ that shares the COLOR feature ‘green’ with the target card by pressing the response button that is labeled ‘2’. (**A**) In a previous study [9], we stratified perseverative errors (PE) by trial-by-trial transitions of selected key cards/corresponding responses. Key card/response transitions are highlighted in light blue (see the framed key cards/response buttons on trial *t*). In the depicted example, re-application of the COLOR rule would result in a PE; potential PE are shown as red response buttons on trial *t*. We compared repetitive PE that implied repeated key cards/responses (right panels) with non-repetitive PE that implied altered key cards/responses (left panels). Repetitive PE comprise the repetition of key cards (incl. prioritized features, i.e., ‘green’ on both trials in the example) and of responses (‘inside-left’ on both trials in the example). Non-repetitive PE comprise the alteration of key cards (incl. prioritized features, i.e., from ‘green’ on trial *t* − 1 to ‘blue’ on trial *t* in the example) and of responses (from ‘inside-left’ on trial *t* − 1 to ‘outside-right’ on trial *t* in the example). The *y*-axis shows time in discrete trial-based units (*t* − 1, *t*). (**B**) ESE refer to the finding that conditional probabilities of repetitive PE (involving key card and response repetitions) are lower than conditional probabilities of non-repetitive PE (involving key card and response alterations). Our initial ESE study [9] was based on a sample of neurology inpatients who were assessed by means of a paper-and-pencil version of the Wisconsin card-sorting task. (**C**) In a follow-up study [10], we replicated ESE in a relatively large sample of young volunteers who were assessed by means of a computerized version of the Wisconsin card-sorting task (cWCST). (**B**,**C**) The *y*-axes show conditional PE probabilities (sample means, inter-individual variabilities).

**Figure 2 brainsci-13-00919-f002:**
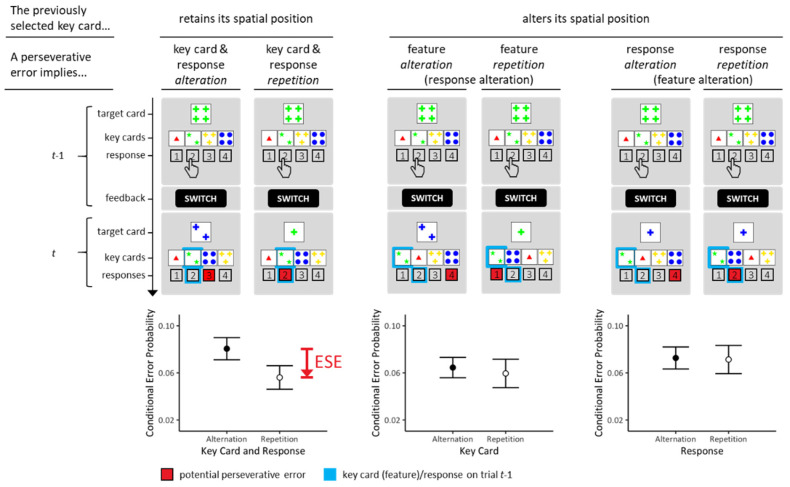
The design (**top panels**) and results (**bottom panels**) of Study 1, which examined non-associative and associative accounts of ESE. (**Top panels**): The allocation of the four key cards to spatial positions was randomly rescheduled on each trial. The main comparison was between those trials *t* on which key cards that were selected on trial *t* − 1 retained their spatial positions on trial *t* (**left column**) and those trials *t* on which key cards that were selected on trial *t* − 1 altered their spatial positions on trial *t* (**central and right columns**). The *y*-axis shows time in discrete trial-based units (*t* − 1, *t*). (**Bottom panels**): The *y*-axes show conditional PE probabilities (sample means, inter-individual variabilities). (**Left column**): ESE were discernible (i.e., repetitive PE were reduced compared to non-repetitive PE) when previously selected key cards re-occurred at their previous spatial positions, thereby replicating the ESE (see also Figure 1; [9,10]). (**Central column**): When previously selected key cards altered their spatial positions, no ESE were discernible when solely the previously prioritized key-card features were repeated (versus altered). (**Right column**): When altered key cards occurred at previously selected spatial positions, no ESE were discernible when the previous responses were repeated (versus altered). Thus, ESE are dependent on the repeated presentation of identical key cards (incl. repetition of prioritized features) at one and the same spatial position. See text for more details.

**Figure 3 brainsci-13-00919-f003:**
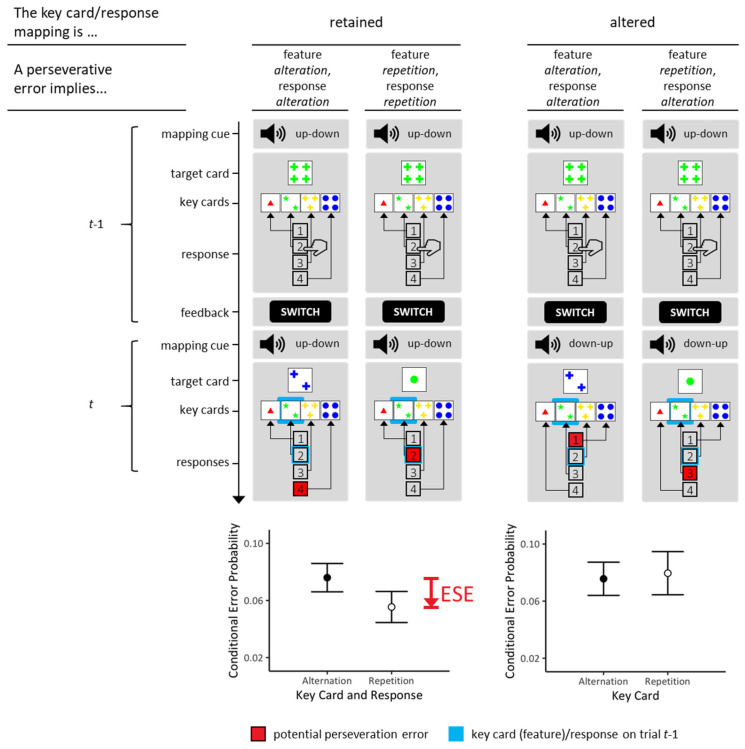
The design (**top panels**) and results (**bottom panels**) of Study 2, which investigated associative accounts of the origin of ESE. As in standard Wisconsin card-sorting tasks, the key cards occupied constant spatial positions. We manipulated the key-card/response mapping on a trial-by-trial basis, with two different mappings. One mapping is the ‘up-down’ mapping (i.e., ‘1 red triangle’ key card maps to response button 1; ‘2 green stars’ key card maps to response button 2; ‘3 yellow crosses’ key card maps to response button 3; ‘4 blue circles’ key card maps to response button 4). The other mapping is the ‘down-up’ mapping (i.e., ‘1 red triangle’ key card maps to response button 4; ‘2 green stars’ key card maps to response button 3; ‘3 yellow crosses’ key card maps to response button 3; ‘4 blue circles’ key card maps to response button 1). (**Left column**): ESE were discernible (i.e., repetitive PE were reduced compared to non-repetitive PE) as long as the S-R mapping remained unchanged across trials, thereby replicating the ESE (see also Figure 1; [9,10]). (**Right column**): No ESE were discernible when the S-R mapping changed across trials. The y-axes show conditional PE probabilities (sample means, inter-individual variabilities). See text for more details.

**Figure 4 brainsci-13-00919-f004:**
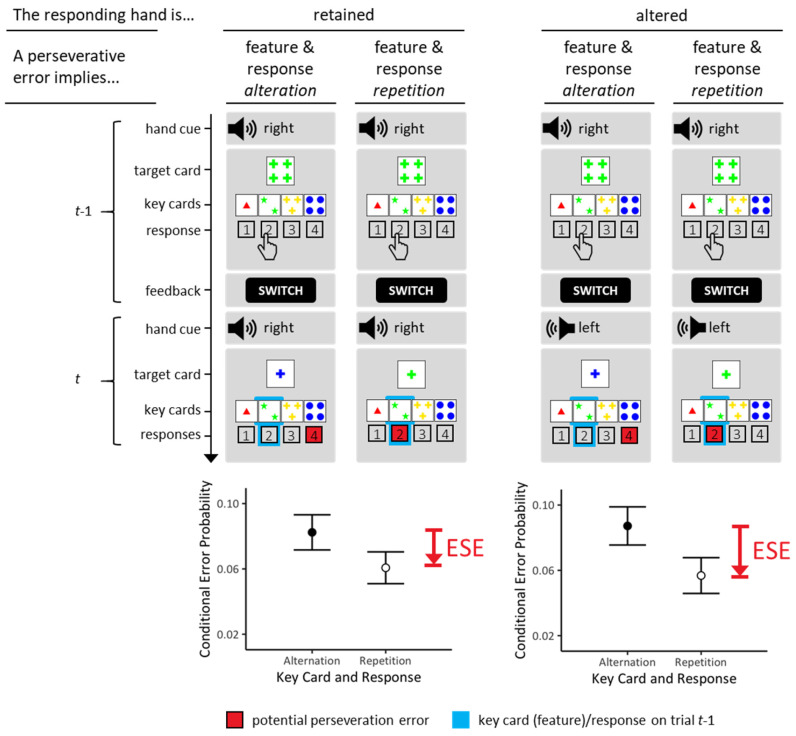
The design (**top panels**) and results (**bottom panels**) of Study 3, which investigated the nature of associative accounts of the origin of ESE. As in standard Wisconsin card-sorting tasks, the key cards occupied constant spatial positions. We manipulated the response/hand mapping on a trial-by-trial basis, with two different mappings. One mapping represented the ‘respond with the right hand’-mapping. The other mapping represented the ‘respond with the left hand’-mapping. (**Left column**): ESE were discernible (i.e., repetitive PE were reduced compared to non-repetitive PE) when the response/hand mapping remained unchanged across trials, thereby replicating the ESE (see also Figure 1; [9,10]). (**Right column**): ESE were discernible (i.e., repetitive PE were reduced compared to non-repetitive PE) even though the response/hand mapping changed across trials, indicating that ESE are not effector specific. Presumably, response codes are specified at a conceptual level such as ‘button 1’ = ‘outside-left’, ‘button 2’ = ‘inside-left’, ‘button 3’ = ‘inside-right’, ‘button 4’ = ‘outside-right’. The *y*-axes show conditional PE probabilities (sample means, inter-individual variabilities). See text for more details.

**Figure 5 brainsci-13-00919-f005:**
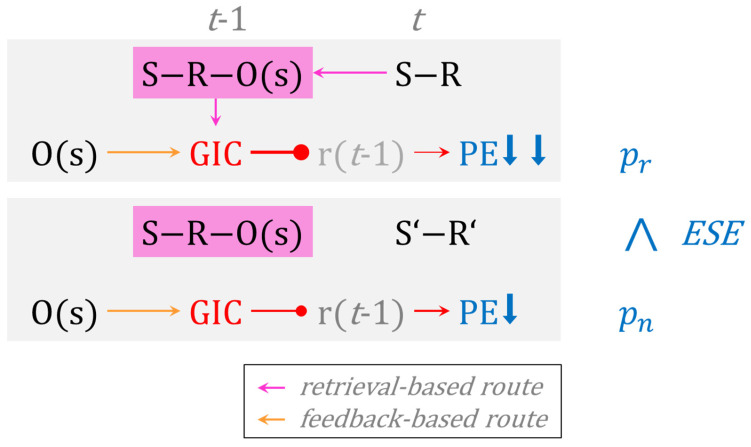
A graphical outline of the goal-directed instrumental control (GIC) model of cognitive perseveration. The left column shows presumed cognitive events (mental representations are shown black; processes are shown in warm colors (orange, pink, red); their observable behavioral expressions are shown in blue; blue arrows show strength of PE decrease) on exemplary single trials (*t* − 1 (affording a rule switch, *t*). The GIC model distinguishes between retrieval-based and feedback-based routes to GIC. Top panel: On repetitive trials, repeating identical S-R conjunctions (S-R) on *t* − 1 and on *t* retrieves goal-directed instrumental memories (pink rectangle). This retrieval brings back to mind that the application of the corresponding rule had just now been disconfirmed via O(s). As an effect of this retrieval-based route to GIC, previously prioritized rules (r(*t* − 1)) are subject to additive (retrieval-based plus feedback-based) inhibition from GIC (shown in red); hence, repetitive PE are strongly suppressed. Bottom panel: On non-repetitive trials, altered S-R conjunctions on *t* − 1 (S-R) and on *t* (S’- R’) do not retrieve goal-directed instrumental memories (pink rectangle). As an effect, previously prioritized rules (r(*t* − 1)) are solely subject to feedback-based inhibition from GIC (shown in red); hence, non-repetitive PE are less strongly suppressed. The right column shows emerging summary statistics of observable PE across multiple trials: Repetitive PE ($p_{r}$) are reduced compared to non-repetitive PE ($p_{n}$), paving the way for what we refer to as conjunctive error-suppression effects (ESE). S, S’ = stimulus (rule-contingent feature); R, R’ = response (spatial code); O(s) = outcome (‘switch’); r(*t* − 1) = prioritized rule on *t* − 1; PE = perseverative error. Arrows as endpoints: activation; circles as endpoints: inhibition. See text for more details.

**Table 1 brainsci-13-00919-t001:** Number of trials (*M*, *SD*) per condition of interest and per participant that could be analyzed in Study 1.

The Previously Selected Key Card …	Retains Its Position		Alters Its Position			
A Perseverative Error Implies…	Key Card and Response Alteration	Key Card and Response Repetition	Feature Alteration	Feature Repetition	Response Alteration	Response Repetition
*M*	44.93	20.20	94.48	18.80	83.33	21.78
*SD*	9.11	4.94	16.15	5.14	15.02	5.52

**Table 2 brainsci-13-00919-t002:** Number of trials (*M*, *SD*) per condition of interest and per participant that could be analyzed in Study 2.

The Key Card/ Response Mapping is…	Retained		Altered	
A Perseverative Error Implies…	Feature and Response Alteration	Feature and Response Repetition	Feature and Response Alteration	Feature Repetition and Response Alteration
*M*	48.25	19.08	25.58	10.30
*SD*	10.27	5.88	5.76	3.33

**Table 3 brainsci-13-00919-t003:** Number of trials (*M*, *SD*) per condition of interest and per participant that could be analyzed in Study 3.

The Responding Hand Is…	Retained		Altered	
A Perseverative Error Implies…	Feature and Response Alteration	Feature and Response Repetition	Feature and Response Alteration	Feature and Response Repetition
*M*	64.75	26.77	64.03	27.65
*SD*	12.43	4.92	18.54	6.95

## Data Availability

The datasets used and/or analyzed for the study are available from the corresponding author upon reasonable request.

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
