# Peer review of "Habits, Goals, and Behavioral Signs of Cognitive Perseveration on Wisconsin Card-Sorting Tasks"

_brainsci, 2023, doi:10.3390/brainsci13060919_

Round 1

Reviewer 1 Report

The manuscript is an interesting neuropsychological paper reporting new insights on the Wisconsin Sorting Card Test, which is a worldwide task used to evaluate prefrontal executive functions. The paper comprises 3 tests and focuses on evaluating the error-suppression effects (ESE). The ESE seems to be an interesting evaluation with a potential clinical role. The paper is well structured and it is clear. I do not have specific concerns about the manuscript. I think it might be considered after a few revisions that might clarify some aspects:

- are the 3 study participants all different?

- have you evaluated participants for mental disorders (for example anorexia nervosa or psychosis) or developmental disorders that might compromise the results? If yes, how? If not, please justify.

- have you evaluated the blindness of the participants to the task? Because if they were university students they might now the Wisconsin.

Author Response

We wish to thank R1 for his/her positive evaluation of our work. Your specific points were addressed in the new paragraph 2.1. (lines 119-130).

Reviewer 2 Report

This is a very interesting article describing a topic that is relevant to the journal. However there are some points that need revision:

At the beginning of their article, the authors should have a broader introduction regarding perseveration not only in this task, but also in other tasks and in different populations (healthy and patients). Therefore, a brief discussion on the following could get the reader who is not involved in neuropsychological research aware of perseveration in general and applied to WCST:

Nagahama, Y., Okina, T., Suzuki, N., Nabatame, H., & Matsuda, M. (2005). The cerebral correlates of different types of perseveration in the Wisconsin Card Sorting Test. Journal of Neurology, Neurosurgery & Psychiatry, 76(2), 169-175.

Giannouli, V. (2013). Number perseveration in healthy subjects: Does prolonged stimulus exposure influence performance on a serial addition task?. Advances in Cognitive Psychology, 9(1), 15.).

Giannouli, V. (2011). Music in a serial repetition task: Is there perseverative behavior. Acta Neuropsychologica, 9, 361-368.

The pictures are hard to follow, so please explain them in the text.

All trials on which participants committed odd errors as well as all trials that followed odd errors were excluded from statistical analyses. Why is that? Please explain  to the reader.

In the methodology section the recruitment of the participants needs to be described in more detail.

Please also state the exclusion criteria.

Is the measure used standardized in your country (validity, reliability)? If not, the authors need to provide more info on that.

In the results section, the tables have to be described in the text. The findings are not clear to the reader.

Finally, the discussion should be more detailed. I understand the importance of Bayesian statistics, but the author(s) do not discuss if any other statistical analyses are appropriate and if no, why. A discussion on what is happening in other countries regarding this issue could further improve the paper.

None. There are minor typos to correct. Overall, it is ok.

Author Response

We wish to thank R2 for his/her positive evaluation of our work.

Concerning broadening the Introduction

We agree with you that the Introduction has a relatively narrow scope, and that the broader context why cognitive perseveration - especially as measured by Wisconsin card-sorting tasks - is of eminent importance to the field, is not presented in depth. However, we think that the manuscript has already a quite exhaustive length due to its layout of three consecutive studies. Readers who are interested in the modulation of behavioral signs of cognitive perseveration may - in their vast majority – already be quite well informed about the relevance of perseveration for understanding functions and dysfunctions of the frontal lobes. Also, since we did not address neural substrates of cognitive perseveration in these studies, an introduction into this topic might be overambitious and even a bit misleading. We think that your suggestion is well-grounded, but it clearly calls for a separate stand-alone paper.            

Explanation of Figures

We also agree that the figures are quite complex. It is for this reason that we spent an awful lot of time and effort not only to optimize their visual appearance, but also their explanation in the figure captions. So, we believe that we already implemented your request to explain the figures in the best way that came to our mind. 

Exclusion of odd errors and post-odd error trials

We added 2 sentences (lines 170-174).   

Recruitment and exclusion crtieria

Your point was addressed in the new paragraph 2.1. (lines 119-130).

Standardization, reliability and and validity of ESE

ESE on Wisconsin card-sorting tasks are a novel phenomenon - until now observed and examined at group levels - that did not yet receive any standardization. In fact, we suggest here how ESE may be measured at individual levels in the Discussion. This individualized operationalization of ESE is of course a prerequisite for the analyses that you suggested (standardization, reliability, validity). In other words, future work is required.    

Tables in the text

The tables are integrated in the text now.

Discussion of Bayesian statistics

You are right in your criticism that we did not defend our decision for Bayesian statistics. We added a short defense of our choice in the revised manuscript (lines 186-198).